behaviour, ecology

predator–prey, noise, bats, foraging behaviour, distraction

**Authors for correspondence:**
Louise C. Allen
e-mail: allenl@wssu.edu
Jesse R. Barber
e-mail: jessebarber@boisestate.edu

†Present address: Department of Biology, University of Florida, Gainesville, FL, USA.

# Noise distracts foraging bats

Louise C. Allen[1], Nickolay I. Hristov[2], Juliette J. Rubin[3,†], Joseph T. Lightsey[1] and Jesse R. Barber[3]

[1]Department of Biological Sciences, Winston-Salem State University, Winston Salem, NC, USA
[2]TERC, Cambridge, MA, USA
[3]Department of Biological Science, Boise State University, Boise, ID, USA

LCA, 0000-0002-0271-4264; JJR, 0000-0002-5143-308X

Predators frequently must detect and localize their prey in challenging environments. Noisy environments have been prevalent across the evolutionary history of predator–prey relationships, but now with increasing anthropogenic activities noise is becoming a more prominent feature of many landscapes. Here, we use the gleaning pallid bat, *Antrozous pallidus*, to investigate the mechanism by which noise disrupts hunting behaviour. Noise can primarily function to *mask*—obscure by spectrally overlapping a cue of interest, or *distract*—occupy an animal's attentional or other cognitive resources. Using band-limited white noise treatments that either overlapped the frequencies of a prey cue or did not overlap this cue, we find evidence that distraction is a primary driver of reduced hunting efficacy in an acoustically mediated predator. Under exposure to both noise types successful prey localization declined by half, search time nearly tripled, and bats used 25% more sonar pulses than when hunting in ambient conditions. Overall, the pallid bat does not seem capable of compensating for environmental noise. These findings have implications for mitigation strategies, specifically the importance of reducing sources of noise on the landscape rather than attempting to reduce the bandwidth of anthropogenic noise.

## 1. Introduction

Predators often have limited strike opportunities such that, while they might live another day after missing dinner, they probably would not live many more [1]. Thus, an evolved excellence at hunting in their environment is paramount. However, environments are variable and predators must often accommodate difficult abiotic factors such as moonlight [2], short-term weather events [3] and noise. For millennia, predators have had to detect and locate prey in a range of naturally noisy environments—windy prairies, rushing rivers—and more recently, in anthropogenically noisy ones—roadsides, gas extraction fields. Although the acoustic environment probably shapes predator distribution and behaviour patterns, it remains an often underappreciated niche axis, to the detriment of conservation planning [4]. Consistently noisy areas can have a filtering effect, where animals remain only if they possess traits that are adaptive for high amplitude backgrounds [5]. Acoustically heterogeneous environments, on the other hand, may drive animals to either avoid these areas during noisy bouts, or behaviourally compensate for information loss or obfuscation if they cannot (reviewed in [6,7]). Those that do not escape or adapt will probably face fitness costs.

While total avoidance of noise might be a preferred strategy for some animals, others might not be able to flee so effectively, due to high site fidelity [8], reticence to pass through fragmented areas [9], or a landscape-scale paucity of quiet spaces from anthropogenic activities [10]. For those that stay, environmental noise can perturb information processing by predators in two main ways: (i) *masking*, or *energetic masking*—the increased difficulty of detecting or discriminating a stimulus of interest due to an alternative source that overlaps the stimulus in spectrum, intensity and time [11]; or (ii) *distraction*—the increased

difficulty of detecting or discriminating a stimulus of interest due to an alternative source that occupies attentional resources and processing power of the organism [12] (although see Dominoni *et al.* [4] for a third, less common mechanism). While studies across taxa, including fish [13,14], birds [15–17] and bats [18–22] have found serious declines in hunting success in noisy environments, the underlying mechanism remains elusive. A deeper, mechanistic understanding can provide insight into the evolutionary biology of how organisms compensate for naturally noisy environments. It can also provide critical direction for mitigation strategies. If predators are most susceptible to masking by some frequencies of noise, band-limiting anthropogenic noise outputs could rescue natural behaviours. If, on the other hand, predators mostly suffer from distraction in noise, it will be more important to simply limit noise on the landscape [4,7,23].

As an acoustically oriented group that comprises 20% of all mammals [24], bats provide a strong model for understanding noise disruption. If given the option, it appears that bats prefer to avoid noisy environments [20] and those that must hunt in noise are less efficient than in quiet conditions (i.e. longer search time to find prey) [19,21]. However, there has yet to be a sufficient, ecologically relevant design to parse the underlying causes of this reduction in efficiency. Using the gleaning greater mouse-eared bat (*Myotis myotis*), Siemers & Schaub [19] found evidence for masking; search time increased in a dose-response fashion with higher background noise levels, but improved somewhat when the noise contained short silent gaps. However, previous work in another taxon (hermit crabs) in noise found a direct relationship between the intensity of a noise stimulus and the crab's distraction level [25]. Thus, the dose-response increase in search time does not clearly distinguish mechanism. Another study found no evidence for masking or distraction in Daubenton's fishing bat (*Myotis daubentonii*); however, the researchers attempted to mask echolocation, which probably did not occur due to insufficient source levels at these higher frequencies. Moreover, noise stimuli were initiated as the bat approached a prey item during foraging bouts, and therefore, a startle response at noise onset could explain the apparent aversive behaviour by bats [18]. Some mechanistic studies have used choice-test paradigms [22,26,27], which provide important information about discrimination, but do not test the bat's ability to detect and localize prey—a critical component of foraging in the wild.

Here, we aim to parse distraction and masking using the gleaning pallid bat (*Antrozous pallidus*) hunting in a flight room using a prey signal emitted from 1 of 21 evenly spaced speakers that we either spectrally overlap or do not spectrally overlap with noise. While all insectivorous bats use echolocation for orientation, bats who passively glean prey—those that often or sometimes hunt by listening for prey-generated sounds on substrates—might be more susceptible to partial or full masking, by low-frequency environmental sounds, such as rivers and traffic, that could spectrally overlap prey footsteps [28]. We predicted that if *masking* were the driving mechanism, we would see a reduction in hunting success and an increase in search time in overlapping noise conditions compared to the ambient and non-overlapping noise condition. If *distraction* were the driving mechanism, we expected to see equal deficits in hunting success and longer search time in both noise treatments in comparison to ambient. Previous work with pallid bats suggests that they

cannot use sonar to image small, stationary prey animals [29], and thereby compensate for any deficiency in their ability to passively glean in masked conditions (as in [22]). However, given that they can change the acoustic parameters of their sonar in a challenging navigation task [30], we predicted that if masked they would alter their rate of sonar emissions in an attempt to image the prey whose passive acoustic cues would be obliterated in the overlapping treatment only. Alternatively, if distracted, we predicted they would alter their sonar in both noise treatments in an attempt to gather more information generally.

## 2. Materials and methods

### (a) Study species and care

We captured five *Antrozous pallidus* bats (3 males, 2 females) using mistnets on a single night in early August 2019 in Ada County, Idaho. All bats were housed in same sex group enclosures, under an inverted light regime (16 h day: 8 h night), and were maintained according to bat care and housing protocols outlined in our IACUC protocol (AC18-007) at Boise State University. To maintain body condition and reinforce training, we flew all bats every day in a dark, anechoic flight room (7.6 m × 6.7 m × 3 m), including non-trial days, and fed bats only during these flight sessions with vitamin-dusted mealworms (*Tenebrio* larvae). We trained all bats under ambient conditions for one month before trials began using the same protocol we followed during data collection. Bats were never exposed to noise treatments until experiments began. While all five bats underwent training, one bat (F) would not reliably land on the speaker playing prey cues under ambient conditions and a second (M) refused to fly in the noise treatments. We could not disentangle these behaviours from prior experience with noisy areas, generalized lab stress, or decreased food motivation thus both bats were dropped from the study before data collection.

### (b) Acoustic stimulus and noise

We inset 21 speakers (Peerless by Tymphany XT25SC90-04) spaced 30.5 cm apart in a 3 × 7 grid in an insulation board (5 × 122 × 244 cm) covered in textured shelf liner so that bats would not slip while landing (figure 1*a,b*). To allow the experimenter to select one speaker from the set of 21 to play the prey signal, we wired all speakers to a switch, connected to an amplifier (Lepai LP-2020TI) and an audio player (Roland R-05). The prey signal was a WAV recording, sampled at 96 kHz, of a cricket walking on a wooden surface covered in butcher paper that we spectrally constrained in SASLab Pro using both a high-pass and low-pass IIR Butterworth filter to arrive at a 4–14 kHz band (figure 1*c–e*). We played this prey cue from one of the 21 inset speakers at approximately 46 dB(Z) (approx. 17 dB(Z) at 30 cm assuming spherical spreading; measured from directly above at 1 cm in a room with a 44 dB(Z) background). During initial training, we exposed all bats to the full-spectrum cricket walking sounds (1.6–17.6 kHz at ± 15 dB, recorded at 10 cm), then gradually limited the bandwidth of the prey cue by contracting the upper and lower frequency bounds until the bat was unable to localize the sound source. We then returned the prey cue to their last localizable frequency bandwidth (4–14 kHz) and used this baseline to build our overlapping (figure 1*d*) and non-overlapping (figure 1*e*) noise treatments.

To create noise treatments that were either overlapping or not overlapping with the prey cue we filtered broadband white noise to 4–14 kHz and 14–24 kHz, respectively, using the IIR Butterworth filter. These treatments are probably received as an equal sensation level by pallid bats [31]. Additionally, based on

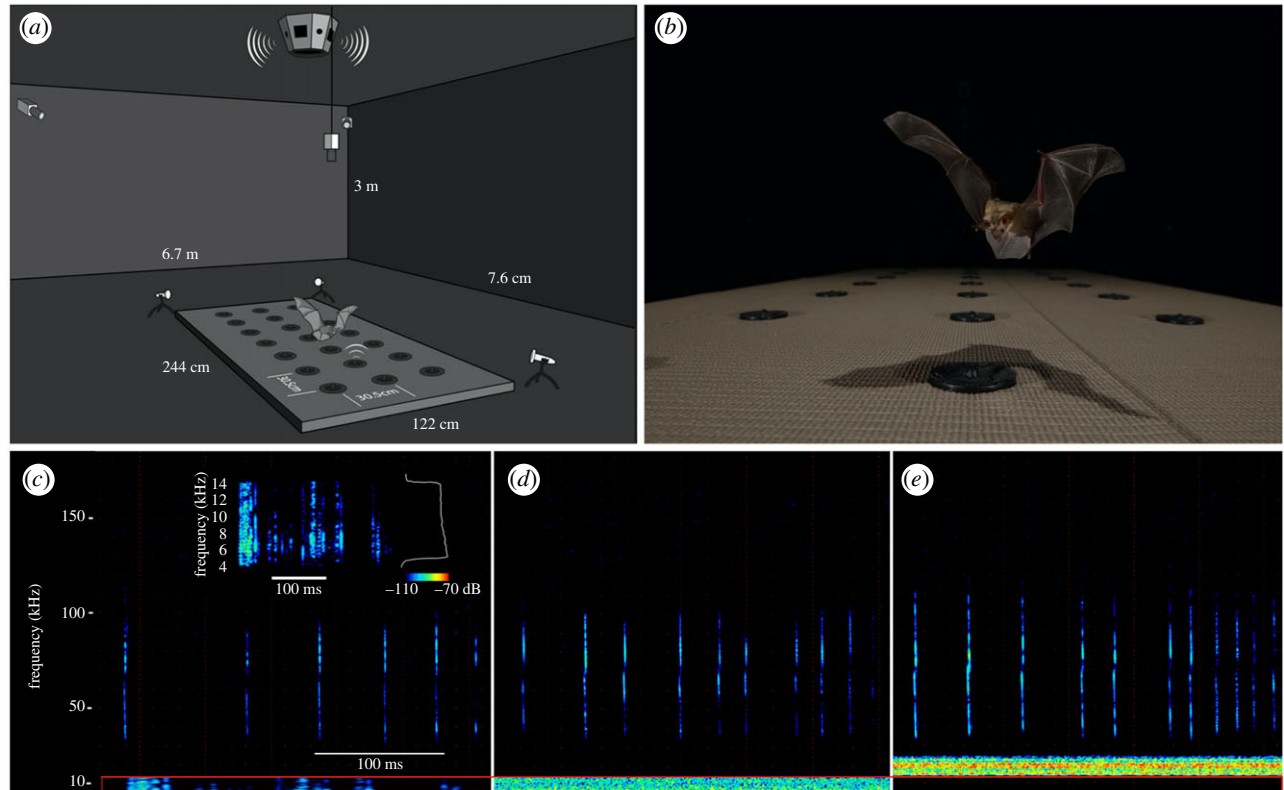

**Figure 1.** (*a,b*) Experimental set-up. (*a*) Bats hunted individually in an anechoic flight room. We broadcast noise treatments from an omnidirectional speaker mounted on the ceiling to parse mechanism of noise disturbance. (*b*) Bats were trained to localize a prey cue playback from 1 of 21 speakers. (*c–e*) Spectrograms (kHz × time; colour indicates intensity, see legend in 1*c* inset) of pallid bat sonar emissions in (*c*) ambient—no noise broadcast, (*d*) overlapping noise—band-limited 4–14 kHz noise that overlapped the spectrum of the prey cue (indicated by the red outlined bar in all three panels), and (*e*) non-overlapping noise—band-limited 14–24 kHz noise that was spectrally distinct from the prey cue and the bat's sonar. The inset panel in (*c*) shows the spectrogram (kHz × time) of the prey signal (4–14 kHz) with its associated power spectrum (kHz × dB) with sound intensity measured in relative dB. Panel (*c*) also includes an overlay of the prey signal and the two spectrograms were combined in Photoshop to visualize the signal that was presented to echolocating bats. The power spectra of the noise stimuli are not presented; we used band-passed white noise with approximately equal energy across frequencies. (Online version in colour.)

estimated critical bands for another bat [32], we think it likely our noise stimuli were perceptually distinct to the bat. Further, the non-overlapping treatment remained below the minimum frequency of pallid bat sonar (approx. 30 kHz) [29]. We broadcast noise treatments from a multi-directional speaker hanging from the ceiling (Octasound SP820A speakers (KDM Electronics Incorporated); approximately 3 m from the ground) using a Roland R05 player and a PRV Audio AD1200.1-2 amplifier, powered by a LiFePO3 (Batteryspace, CA, USA) battery. The levels were overlapping = 50 dB(Z) and non-overlapping = 50 dB(Z), measured 30 cm from the board. During our control ambient trials, the background noise in the room was 45.5 dB(Z) and 33.5 dB(A). We measured all sound levels with a Brüel & Kjær (B&K) 2610 measuring amplifier and B&K ¼' microphone type 4939-A-011 (grid off).

## (c) Experimental protocol

We randomized (with replacement) the speaker from which we broadcast the prey cue (1–21) and the acoustic environment treatment (ambient, overlapping and non-overlapping) for each trial, with an equal number of trials for each treatment per night per bat (up to 15 trials per bat). Our experimental trials spanned 10 consecutive nights, 6–15 October 2019. We also randomized the presentation of a single, freshly killed mealworm reward atop the prey cue speaker to reinforce correct speaker choices, with an average presentation rate of 1 mealworm absent to 9 mealworms present. We subsequently built mealworm presence/absence into our models to control for possible echolocation or olfactory cues that the bat might detect when the mealworm was present.

A single experimenter controlled all parts of the experimental protocol from within the flight room, while a second ran the recording equipment from a separate control room. For each trial, the experimenter placed a prey reward on the correct speaker, in addition to pretending to place the prey reward on other speakers in case the bats were tracking this movement. The same procedure was also carried out in non-rewarded trials. The experimenter then sat against a wall, away from the board, and began the noise file (overlapping, non-overlapping or no noise file for ambient). After 15 s passed (for acclimation), the experimenter began the prey signal and simultaneously triggered a brief (less than 10 ms) visible and audible flash to synchronize audio and video recordings. A trial ended when the bat either landed on the board (i.e. made a foraging attempt) or after 60 s from prey signal onset.

## (d) Video and audio capture/analysis

Over 10 nights of data collection, we recorded more than 440 video and audio files on three bats' behaviour in a dark flight room. We used three Basler ace 2 cameras (1920 × 1080, 100 frames per second), driven by STREAMPIX 8 software on a Norpix system via USB 3.0 connection, and synchronized using a customized sync box. We synchronized video and audio footage using an outgoing trigger pulse from StreamPix to a four-channel Avisoft Ultra-SoundGate 416H (sampling at 250 kHz), recording from four ultrasonic Avisoft microphones to a desktop computer running AVISOFT RECORDER software. We placed three microphones (CM16 ± 3 dB(Z), 20–140 kHz) each at a corner of the board and angled them slightly upwards. We also placed a fourth, omnidirectional

microphone (Avisoft Electret microphone, ±9 dB(Z), 30–120 kHz) on the ceiling and pointed it directly at the board. We analysed video files to measure attempt, success and search time. We defined attempt as the bat landing on the board to make a foraging attempt within 60 s of the prey signal; success as the bat successfully landing on the correct speaker; and search time as the amount of time it took for the bat to make a speaker selection after the signal start. We selected the microphone with the best signal-to-noise ratio to extract echolocation parameters in SASLab Pro software. These parameters included spectral (dominant frequency) and temporal (average interpulse interval (IPI)) characteristics. When faced with a challenging task, many bat species will increase their rate of echolocation calls to gather more information, effectively reducing the IPI [22,33], or emphasize different harmonics of their call [34]. Thus, both IPI and dominant frequency can serve as metrics of the bat's perception of task difficulty. Additionally, a shift in dominant frequency could be indicative of a Lombard effect, where an animal elevates the amplitude (and frequency as a by-product) of their vocalization in an attempt to accommodate background noise [35].

We analysed all audio files from the first and final three nights of the experiment where the bat made an attempt (142 files; ambient = 54, overlapping = 38, non-overlapping = 50). From each file, we analysed the call sequence preceding an attack, that is, the 1-second window leading up to the bat landing on the board (coded as an attempt), using Avisoft SASLab Pro software (Hamming window, 1024 fast Fourier transform, Threshold:50). We measured dominant frequency using the power spectrum tool (averaged; Hann window, 1024 fast Fourier transform). For files containing either of the two noise treatments, we excluded all low-frequency peaks from our power spectrum window using the zoom tool. To measure average IPI, we filtered noise treatment files with a time-domain FIR filter at 20 kHz for the overlapping noise and 25 kHz for the non-overlapping noise, and then averaged the IPI values extracted by the pulse train analysis tool in SASLab Pro (see electronic supplementary material; [36]).

## (e) Statistical analysis and model descriptions

To test our predictions regarding the effects of noise on foraging behaviour in pallid bats, we used a series of generalized linear mixed-effects models with different distribution families and link functions based on data type, using the package lme4 [37] in R [38]. Our response variables were attempt, success, search time, average IPI and dominant frequency. We used binomial distributions with logit links for all attempt and success models, gamma distributions for search time models, and linear models for IPI and dominant frequency models. We checked residual plots with the package DHARMa [39]. To account for individual bat variability and repeated sampling of individuals, we used bat ID as a random intercept in all models except attempt, as attempt did not converge within this model structure. While bat ID had limited levels (three bats) [40], we retained this random effect in our models to control for the repeated-measures design. For our success, search time, average IPI models and dominant frequency models, we only included trials where the bat made an attempt. We used ggplot2 [41] and sjPlot [42] to build all figures.

To determine whether having a prey reward on the board affected the outcome of the trial, according to any one of our behavioural parameters, we modelled each of these response variables against an interaction term of treatment and mealworm presence/absence (see electronic supplementary material; [36]). To test for the effect of experience on trial outcome, we re-built all models using the experimental night as the exclusive fixed effect and bat as a random intercept. We found no significant effect of experimental night or prey reward presence/absence, and therefore, we did not include these parameters in our final

models, except for the attempt model, where we modelled night as an interaction term.

## 3. Results

Bats demonstrated greater difficulty foraging in noise, regardless of whether the noise spectrally overlapped the prey signal. In both noise types, bats were less likely to make a foraging attempt (probability of attempting: ambient mean = 1.0; overlapping mean = 0.79, 95% CI = 0.71–0.87; non-overlapping mean = 0.79, CI = 0.72–0.87) (figure 2*a*). Unique among the response variables, there was a temporal component to this behaviour—bats increased their attempt rate in noise as experimental nights progressed (slope: overlapping = 0.60, non-overlapping = 0.58), while their attempts remained consistently near 100% in the ambient condition (slope: ambient = $2.56 \times 10^{-07}$) (figure 2*a*). Bats were also less successful at correctly localizing prey in noise (probability of success: ambient mean = 0.94, CI = 0.89–0.98; overlapping mean = 0.38, CI = 0.28–0.51; non-overlapping mean = 0.33, CI = 0.24–0.46) (figure 2*b*). Similarly, bats were equally inefficient at making their predatory strike in either noise type, compared with ambient conditions (search time (s): ambient mean = 4.08, CI = 3.29–5.08; overlapping mean = 15.62, CI = 12.40–19.67; non-overlapping mean = 14.95, CI = 11.90–18.80) (figure 2*c*). Overall, foraging deficits were nearly identical between the two noise treatments. Additionally, bats made more use of sonar in noise, emitting significantly more pulses in the 1000 ms before landing for prey capture (average IPI: ambient mean = 94.9, CI = 76.3–113.4; overlapping mean = 76.3, CI = 58.6–93.9; non-overlapping mean = 74.5, CI = 56.2–92.9) (figure 2*d*). They did not, however, alter the dominant frequency of their calls in noise, as they consistently emphasized the second harmonic in all treatments (dominant frequency: ambient mean = 79, CI = 77.5–80.4; overlapping mean = 79.4, CI = 77.8–80.9; non-overlapping mean = 77.8, CI = 76.4–79.2) (see electronic supplementary material) [36].

## 4. Discussion

Our experiments provide evidence that foraging pallid bats are distracted by noise. All three bats exhibited similar deficits in hunting accuracy and efficiency in an acoustic environment that spectrally overlapped a prey cue, as well as one that was spectrally distinct. By limiting the frequency bandwidth of prey walking sounds, we were able to create clear on-band and off-band tests of the effects of noise. This is different from previous studies that have used full-spectrum prey cues as the stimulus [19–22] and gave us more control over our overlapping noise treatment. We predicted that if masking were the mechanism underlying the deleterious effects of noise on hunting, bats would be less successful in localizing their prey and would take longer to do so in overlapping noise than they would in non-overlapping or ambient treatments. We instead found support for our distraction hypothesis as bats suffered a similar drop in successful foraging attempts and foraging efficiency in both noise conditions—providing evidence that distraction is a major mechanism underpinning noise-mediated hunting deficits.

Further evidence for distraction comes from our recordings of bat sonar. Bats increased their echolocation call rate similarly in both noise conditions, yet this change in sonar behaviour did not correlate with improved hunting success in

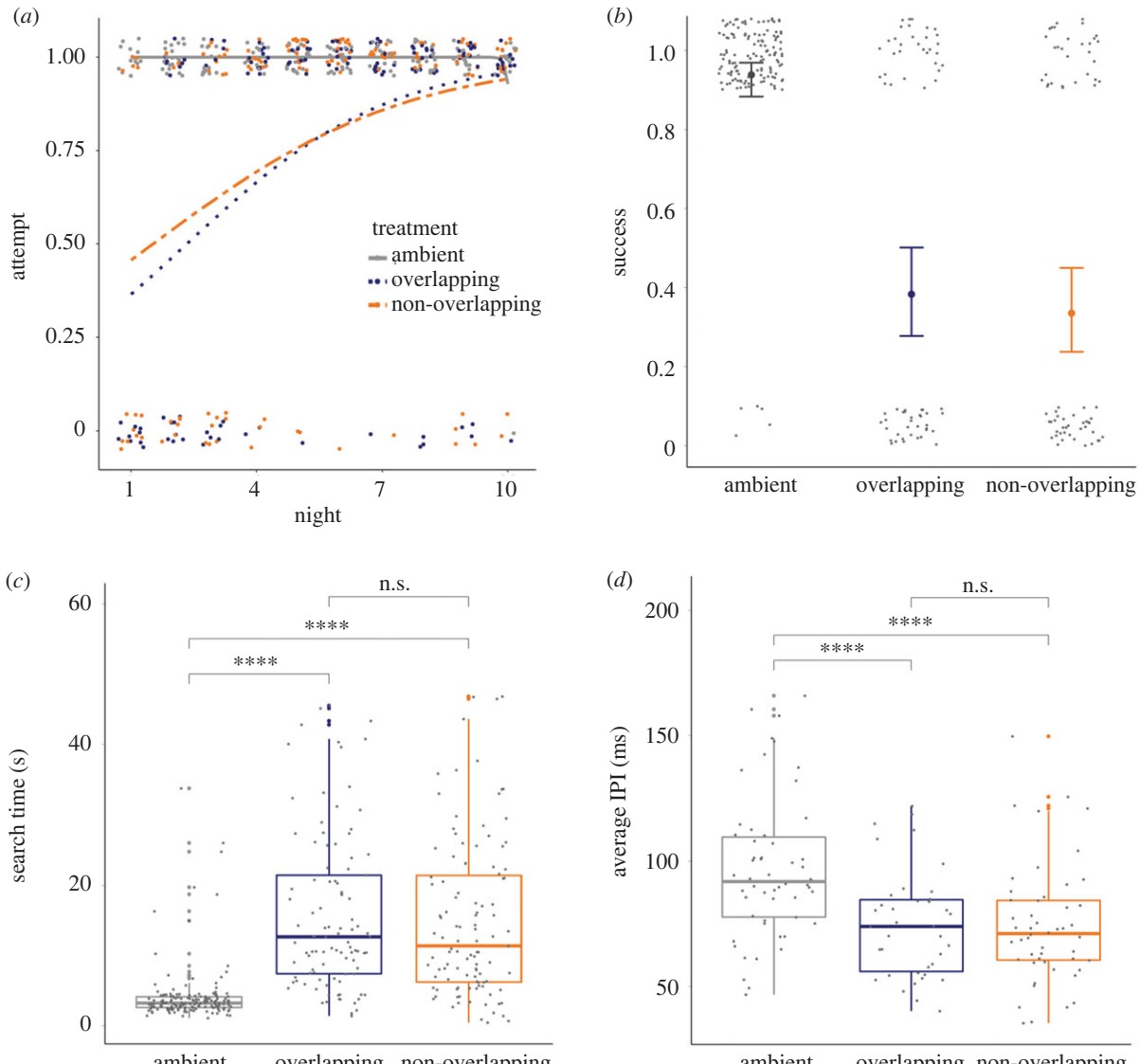

**Figure 2.** Bats demonstrate altered hunting behaviour in noise, regardless of whether noise overlaps the prey cue. In comparison with ambient noise conditions, bats hunting in noise treatments that either spectrally overlap or do not overlap (non-overlapping) a prey cue (*a*) show increased reticence initially to hunt in noise, but over time are more willing to make a foraging attempt, (*b*) are nearly half as successful, (*c*) take almost three times longer to make a foraging attempt, and (*d*) use more sonar pulses (i.e. decrease average IPI). Together, these results are indicative of a distracting effect of noise. *Attempt:* ambient $n = 159$, overlapping $n = 136$, non-overlapping $n = 143$. *Success* and *search time:* ambient $n = 158$, overlapping $n = 101$, non-overlapping $n = 107$, *IPI* ambient $n = 54$, overlapping $n = 38$, non-overlapping $n = 50$. Note that in (*a*–*d*), each data point represents an individual trial, (*b*) portrays the median and 95% CI, (*c*) and (*d*) boxes display the medians, 25th and 75th percentiles, and the whiskers indicate the 5th and 95th percentiles. (Online version in colour.)

noise. This suggests that while bats attempted to gather more information about their environment to make foraging decisions in noise, they perceived both acoustic environments as equally challenging. Despite the congruence of evidence between hunting behaviour and sonar emissions that we present here, it is important to point out that individuals can respond to environmental disturbances differently [26] and that additional work is necessary with more bats and different species.

These results are in accordance with those of a previous laboratory study with pallid bats that found a roughly equal degradation of hunting efficiency in noise, regardless of noise type or sound intensity, leading the authors to hypothesize that distraction was the likely underlying mechanism [21]. Our study did not test different sound levels of the overlapping and non-overlapping noise treatments, and thus, we cannot know whether the pallid bat in this

paradigm would have responded in a dose–response fashion to increasing levels, as was found in the greater mouse-eared bat [19], or whether we would have found consistent deficits across sound intensities, as in Bunkley & Barber [21]. It is possible that the mechanism is different between these two species and that the greater mouse-eared bat did experience a masking effect from noise as Siemers & Schaub proposed [19]. This could have been driven by higher sound levels than those used in our study (although we cannot be sure of this, as the distance from road was reported, rather than decibels). To definitively understand if a dose-response reaction to noise is indicative of mechanism, future work must employ an overlapping/non-overlapping paradigm [18] (this study) coupled with a gradient of treatment sound levels.

A complete understanding of listening in noise requires comprehensive data on the auditory system itself. In general,

auditory systems are binned into overlapping band-pass filters—critical bands—that are indicative of which frequencies might mask a signal. The bandwidths of the auditory filters in pallid bats are unknown. One of only a few such studies in bats [32], using the greater spear-nosed bat (*Phyllostomus hastatus*), a bat that also consumes large insect prey [43], used an indirect approach to measure critical bandwidths and found a span from approximately 0.2–3 kHz over the frequency range of our non-overlapping noise treatment. Given the upward-biased asymmetry of auditory filters in mammals [44], it seems unlikely that this noise treatment would have had any substantial masking effect on the prey signal. We also think it is unlikely that our non-overlapping noise treatment masked echoes from pallid bat sonar emissions as there was a greater than 6 kHz separation between the noise and the minimum frequency of pallid bat sonar (approx. 30 kHz [29]). A more complete set of behavioural audiograms, critical ratios and critical bands are essential for discerning the differing effects of noise on auditory predators more broadly.

Non-auditory system-related processes are also important to consider. For instance, it is possible that simultaneously while taking up attentional resources, our noise playbacks provoked a stress response. Similarly to the aversion response found in Daubenton's fishing bat [18] and the greater mouse-eared bat [20], our bats showed reduced foraging effort in both noise treatments compared with ambient initially, but gradually became fully willing to hunt over time. Different components of animal behaviours can be associated with differential habituation rates (i.e. a decline in response to a repeated stimulus) [45]; however, as bats did not improve in hunting efficiency or accuracy over the course of the experiment, nor did they in similar previous work [21], we think it unlikely, but not improbable, that multi-component habituation was a primary factor driving our results. A robust test of aversion will require a focus on the physiological responses of bats hunting in noise—such as heart rate telemetry—to assess the state of bats under these conditions [46]. In terms of hunting performance, we see stress as separate from the acoustically based mechanisms of masking and distraction that probably acts as a mediator but not a driver of foraging deficits.

Our findings supporting distraction as a primary mechanism driving hunting deficits have important implications. Distraction does not depend on the overlap in properties between the noise and the stimulus. Thus, distraction can occur within and across sensory modalities. Future work should focus on higher-level cognitive processes, such as spatial orientation [47] and memory retrieval [48]. This work would benefit from using the human literature as a guide to form hypotheses for animals. For instance, data indicate that noise interferes with learning and problem-solving capacity in humans, probably due to distraction or related phenomena [49].

Our behavioural results provide further evidence that the acoustic environment is an underappreciated niche axis [50] that likely constrains bat habitat suitability. Further, our results do not bode well for the long-term persistence of acoustic predators such as the pallid bat. It seems that spectrally filtering anthropogenic noise will not mitigate the costs to wildlife, but that instead we must reduce the acoustic footprint of human activity. Moreover, over the course of 10 experimental nights of hunting in noise, our bats did not demonstrate an ability to acclimate to this environment. That is, while they did show increased willingness to hunt in the noise over time, they did not show an increase in efficacy at hunting in noise. Instead, we found that their probability of successfully localizing their prey dropped from 0.94 (CI = 0.89–0.98) in ambient conditions to approximately 0.35 (CI = approximately 0.25–0.50) in both noise treatments—a dismal outcome for a hunter trying to survive. Moreover, our estimation of successful prey localization in noise was possibly an overestimation, as they only had 21 possible prey locations to choose from and an opportunity to make multiple flight passes, while a foraging task in the wild is likely far more disperse. Many bats probably avoid noisy areas rather than suffer these foraging costs [18,20,51], but with ever-expanding cities, roadways and energy extraction fields, it is possible that at some point, acoustically oriented predators simply would not be able to flee far enough [4,10,52]. We posit that many animals suffer a similar effect of distraction in noise and suggest that the weight of scientific evidence is becoming increasingly clear—it is time to pay careful attention to noise management.

**Ethics.** This work was approved by the Boise State Institutional Animal Care and Use Committee (006-AC18-007).

**Data accessibility.** R scripts and data sheets are available from the Dryad Digital Repository: https://doi.org/10.5061/dryad.cz8w9gj27 [53].

**Authors' contributions.** L.C.A. and J.R.B. designed the study. N.I.H. and L.C.A. trained the bats. L.C.A., N.I.H., J.J.R. and J.T.L. collected the data. J.J.R. led data analysis with input from all authors. L.C.A., J.J.R. and J.R.B. wrote the first draft of the manuscript and all authors contributed to the final version.

**Competing interests.** We declare we have no competing interests.

**Funding.** This work was funded by NSF grants to Louise Allen (HBCU RIA 1800687 and AISL 1514766) and Jesse Barber (DEB 1556177).

**Acknowledgements.** We thank Kari Dawson, Lakhia Fuller, Cecelia Miller, Bailey Taylor and Brandt Quirk-Royal for assistance with bat training and data collection; Mark Bee and Dylan Gomes for input on earlier drafts; and Chris Tullar for the illustration in figure 1a. Two anonymous reviewers greatly strengthened our manuscript. We dedicate this paper to Björn Siemers and his foundational work on bats hunting in noise.

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
