## [Reviewer comments · Proceedings of the Royal Society B: Biological Sciences]

Review History

RSPB-2020-2689.R0 (Original submission)

Review form: Reviewer 1

Recommendation

Accept with minor revision (please list in comments)

Scientific importance: Is the manuscript an original and important contribution to its field?
Excellent

General interest: Is the paper of sufficient general interest?
Excellent

Quality of the paper: Is the overall quality of the paper suitable?
Excellent

Is the length of the paper justified?
Yes

Should the paper be seen by a specialist statistical reviewer?
No

Do you have any concerns about statistical analyses in this paper? If so, please specify them explicitly in your report.

No

It is a condition of publication that authors make their supporting data, code and materials available - either as supplementary material or hosted in an external repository. Please rate, if applicable, the supporting data on the following criteria.

Is it accessible?

N/A

Is it clear?

N/A

Is it adequate?

N/A

Do you have any ethical concerns with this paper?

No

Comments to the Author

This is an elegant experiment designed to parse out the mechanism by which noise impedes foraging behavior in bats. While a handful of studies have documented a detriment in bat foraging ability in the presence of noise, none have clearly identified the mechanism underlying this decrease in performance. When confronted with noisy surroundings, do bats show reduced foraging performance because they are distracted, or because target sounds are masked?

The study makes a compelling argument and is an excellent contribution to this growing field. My only real concern is the low sample size. The researchers started with five individuals but were unable to proceed through the full experiment with two of them, leaving a sample size of three. While I find the results very clear (bats showed similar detriment in foraging ability in the presence of both bandwidths of noise, suggesting that it is distraction, not masking, that is impeding foraging performance), I think it is important to stress that individuals can respond to noise (or any environmental disturbance) in different ways, as is nicely evidenced in one of the studies this paper cites (Gomes & Goerlitz 2020). Variation in response is expected in animal behavior, and results from a small number of individuals should be interpreted carefully. I would recommend discussing this sample size limitation explicitly in the discussion in light of Gomes & Goerlitz 2020, perhaps stressing the need for further investigation with more individuals and more species. I think this is a very strong study and an important contribution to the field. But I think some caution should be introduced in the interpretation of the results.

Gomes DGE, Goerlitz HR. 2020 Individual differences show that only some bats can cope with noise-induced masking and distraction. bioRxiv.

I have listed some minor comments below:

Line 89-90: I disagree with this definition of gleaning bats: 'those that listen for prey-generated sounds on substrates.' To my understanding, gleaning simply refers to the foraging technique of capturing prey off of substrate (e.g., foliage, the ground). The sensory mode by which gleaning takes place is often attending to prey-generated sounds ('passive gleaning'). But other sensory modalities can be used too. Active gleaners, for example, can actively produce echolocation calls to find silent, motionless prey resting on leaves in the forest understory (e.g., Geipel et al 2013, Proc Roy Soc: Perception of silent and motionless prey on vegetation by echolocation in the gleaning bat, *Micronycteris microtis*). I'd recommend rewording this sentence a bit.

Line 98: more explanation is needed here to make the meaning clear to the reader: 'Previous work with pallid bats suggests that they cannot use sonar to image small, stationary prey animals, and thereby rescue their foraging behaviors in masked conditions'. Describe in more detail what is meant by 'rescue' and how this rescue is accomplished.

Line 101: add 'we predicted' as follows: 'Alternatively, if distracted we predicted they...'

Line 132: Referring to this line in the methods, 'During initial training, we exposed all bats to the full-spectrum cricket walking sounds', what frequency range was this?

Lines 123-133: To limit the bandwidth, did you keep the lower frequency steady but gradually reduce the upper frequency? Perhaps good to specify this here.

Line 137: change 'signal' to 'cue'

Line 189: change 'cries' to 'calls'

Line 283: I would be more cautious in the classification of the foraging mode of *Phyllostomus hastatus*. While *P. hastatus* does indeed consume many insect prey (as well as flexibly and opportunistically consuming small vertebrates, fruits and flowers), to my knowledge it has not been shown how it catches its insect prey. And while it likely gleans prey from substrates, it is unclear whether it uses prey-produced sounds to do so. More studies are needed on this bat... Maybe change 'a bat that also gleans prey' (line 283) to 'a bat that also consumes large insects'.

Line 294: It would be nice to have more of a discussion of what might be going on during distraction. The authors elude to a reduction in attention or other cognitive processes, and a decrease in processing power. Can you go into a little more depth on what attentional/cognitive processes might be impeded in distraction? What studies have looked at this and what might be done next to investigate these processes in bats?

Line 311: What was the hunting efficacy of the bats without noise? Maybe good to add that here to give perspective (e.g., 'their probability of successfully localizing their prey dropped from XX to approximately 0.35')

In the references, in line 343 italicize '*Canis lupis*'; in line 389 italicize '*Antrozous pallidus*'; in line 403 italicize '*Megaderma lyra*'

In sum, these are minor points that should strengthen an already very strong manuscript, which serves as an excellent contribution to the growing field of the effects of anthropogenic noise on predator foraging behavior.

Review form: Reviewer 2

Recommendation

Accept with minor revision (please list in comments)

Scientific importance: Is the manuscript an original and important contribution to its field?

Good

General interest: Is the paper of sufficient general interest?

Good

Quality of the paper: Is the overall quality of the paper suitable?

Good

Is the length of the paper justified?

Yes

Should the paper be seen by a specialist statistical reviewer?

No

Do you have any concerns about statistical analyses in this paper? If so, please specify them explicitly in your report.

No

It is a condition of publication that authors make their supporting data, code and materials available - either as supplementary material or hosted in an external repository. Please rate, if applicable, the supporting data on the following criteria.

Is it accessible?

Yes

Is it clear?

Yes

Is it adequate?

Yes

Do you have any ethical concerns with this paper?

No

Comments to the Author

Passive gleaning bats that rely on prey generated sounds show a decline in hunting success when foraging in noisy environments. The driving mechanism for this decline could either be masking of the prey generated sound by spectral overlap, or distraction of the bat by drawing its attention from the detection and localization task. The aim of this study was to elucidate which mechanism may explain the decline in the bats' foraging performance. In a well-designed experimental approach, the authors trained pallid bats to land on one of 21 speakers broadcasting band-limited rustling noises of insect prey and measured their foraging attempt, foraging success and foraging time in ambient condition and two bandpass limited noise conditions of which one condition spectrally overlapped with the prey cue and the other did not overlap with the cue. The results show a decline in hunting success, an increase in search time and at least in the beginning of the experiment a reduced attempt to land on the speakers in both noise condition, supporting the distraction hypothesis. In the echolocation behavior bats increased the repetition rate under noise condition, although this had no effect on foraging success. Overall this is a thoroughly conducted experiment which indicates that distraction and not exclusively masking is a primary driver for the reduced foraging performance of passive gleaning bats in noisy environments. This work complements the literature on ambient noise and its effect on foraging success in gleaners and is of high scientific value as it delivers important knowledge on the underlying mechanism. Therefore, this manuscript should be published with some minor revisions (see below).

Minor comments

96f: "Previous work with pallid bat (...), and thereby rescue their foraging behaviors in masked situations" What exactly do you mean? I think this needs a bit more explanation

130: intensity of prey cue is given for 1cm above the speaker, but noise levels are given for the bats' foraging height at 30 cm. Intensity levels should refer to the same distance.

148: Was the background noise in the ambient condition measured at the same position as in the noise condition? The background noise is almost as loud as your prey cue and your noise conditions are only 4.5 dB above the ambient noise level. What was the spectrum of your background noise? I suggest to introduce an own figure showing the sonogram and power

spectrum of your three conditions and of the prey cues. The spectrograms in Fig. 1c-e are too small.

195ff: how many calls from how many trials were analysed?

Fig.2: colours, especially yellow, are hardly visible in the printed version. Lines are too narrow and dots too small.

2a and b: I guess, dots indicate single trials. I find this confusing as they are plotted in a graph with a continuous y- axis. At least, it should be mentioned in the figure legend.

2b: y- axis labeling: why "predicted" success? success is clearly defined in the method section as successfully landing on the correct speaker.

288: "We also think it is unlikely" instead of "We also think it unlikely is..."

In the result section you report data on the echolocation behavior (higher repetition rate, no change in peak frequency). These results are not discussed. How do you explain these results? A short paragraph on echolocation behavior should be added in the discussion.

386: Ostwald instead of Ostwalk; J. Exp. Biol. 211, 3174-3180 instead of 315-324

Decision letter (RSPB-2020-2689.R0)

21-Dec-2020

Dear Dr Allen:

Your manuscript has now been peer reviewed and the reviews have been assessed by an Associate Editor. As you will see, the reviewers and the AE are all enthusiastic about your manuscript, as am I, but have raised some concerns that we would like to invite you to revise your manuscript to address. I will not repeat their reviews here, but do note in particular that I agree that it is important to directly address your sample size limitation and how it might impact interpretation of your results. Please also be sure that your data and R scripts are available on Dryad. The reviewers' comments (not including confidential comments to the Editor) and the comments from the Associate Editor are included at the end of this email for your reference.

Research ethics:

Use of animals and field studies:

It is a condition of publication that you make available the data and research materials supporting the results in the article. Please see our Data Sharing Policies (<https://royalsociety.org/journals/authors/author-guidelines/#data>). Datasets should be deposited in an appropriate publicly available repository and details of the associated accession number, link or DOI to the datasets must be included in the Data Accessibility section of the article (<https://royalsociety.org/journals/ethics-policies/data-sharing-mining/>). Reference(s) to datasets should also be included in the reference list of the article with DOIs (where available).

Please submit a copy of your revised paper within three weeks. If we do not hear from you within this time your manuscript will be rejected. If you are unable to meet this deadline please let us know as soon as possible, as we may be able to grant a short extension.

Best wishes,
Dr Sarah Brosnan
Editor, Proceedings B
mailto: proceedingsb@royalsociety.org

Associate Editor
Board Member: 1
Comments to Author:

Thank you for submitting your manuscript to Proceedings B. We have now received two reviews of your manuscript, both of which express enthusiasm for the elegantly simple experimental design and the broad interest of addressing the mechanism by which acoustically hunting animals are influenced by noise. Both reviewers also provide many valuable comments and suggestions that will improve the clarity of the methods and results in a future version of the manuscript. Reviewer 1, however, expresses concern about the very low sample size and the fact that this is not directly addressed in the manuscript. To be considered for publication in Proceedings B, a revised manuscript would need to effectively address this concern.

Reviewer(s)' Comments to Author:

Referee: 1

Comments to the Author(s)

This is an elegant experiment designed to parse out the mechanism by which noise impedes foraging behavior in bats. While a handful of studies have documented a detriment in bat foraging ability in the presence of noise, none have clearly identified the mechanism underlying this decrease in performance. When confronted with noisy surroundings, do bats show reduced foraging performance because they are distracted, or because target sounds are masked?

The study makes a compelling argument and is an excellent contribution to this growing field. My only real concern is the low sample size. The researchers started with five individuals but were unable to proceed through the full experiment with two of them, leaving a sample size of three. While I find the results very clear (bats showed similar detriment in foraging ability in the presence of both bandwidths of noise, suggesting that it is distraction, not masking, that is impeding foraging performance), I think it is important to stress that individuals can respond to noise (or any environmental disturbance) in different ways, as is nicely evidenced in one of the studies this paper cites (Gomes & Goerlitz 2020). Variation in response is expected in animal behavior, and results from a small number of individuals should be interpreted carefully. I would recommend discussing this sample size limitation explicitly in the discussion in light of Gomes & Goerlitz 2020, perhaps stressing the need for further investigation with more individuals and more species. I think this is a very strong study and an important contribution to the field. But I think some caution should be introduced in the interpretation of the results.

Gomes DGE, Goerlitz HR. 2020 Individual differences show that only some bats can cope with noise-induced masking and distraction. bioRxiv.

I have listed some minor comments below:

Line 89-90: I disagree with this definition of gleaning bats: 'those that listen for prey-generated sounds on substrates.' To my understanding, gleaning simply refers to the foraging technique of capturing prey off of substrate (e.g., foliage, the ground). The sensory mode by which gleaning takes place is often attending to prey-generated sounds ('passive gleaning'). But other sensory modalities can be used too. Active gleaners, for example, can actively produce echolocation calls to find silent, motionless prey resting on leaves in the forest understory (e.g., Geipel et al 2013, Proc Roy Soc: Perception of silent and motionless prey on vegetation by echolocation in the gleaning bat, *Micronycteris microtis*). I'd recommend rewording this sentence a bit.

Line 98: more explanation is needed here to make the meaning clear to the reader: 'Previous work with pallid bats suggests that they cannot use sonar to image small, stationary prey animals, and thereby rescue their foraging behaviors in masked conditions'. Describe in more detail what is meant by 'rescue' and how this rescue is accomplished.

Line 101: add 'we predicted' as follows: 'Alternatively, if distracted we predicted they...'

Line 132: Referring to this line in the methods, 'During initial training, we exposed all bats to the full-spectrum cricket walking sounds', what frequency range was this?

Lines 123-133: To limit the bandwidth, did you keep the lower frequency steady but gradually reduce the upper frequency? Perhaps good to specify this here.

Line 137: change 'signal' to 'cue'

Line 189: change 'cries' to 'calls'

Line 283: I would be more cautious in the classification of the foraging mode of *Phyllostomus hastatus*. While *P. hastatus* does indeed consume many insect prey (as well as flexibly and opportunistically consuming small vertebrates, fruits and flowers), to my knowledge it has not been shown how it catches its insect prey. And while it likely gleans prey from substrates, it is unclear whether it uses prey-produced sounds to do so. More studies are needed on this bat... Maybe change 'a bat that also gleans prey' (line 283) to 'a bat that also consumes large insects'.

Line 294: It would be nice to have more of a discussion of what might be going on during distraction. The authors elude to a reduction in attention or other cognitive processes, and a decrease in processing power. Can you go into a little more depth on what attentional/cognitive processes might be impeded in distraction? What studies have looked at this and what might be done next to investigate these processes in bats?

Line 311: What was the hunting efficacy of the bats without noise? Maybe good to add that here to give perspective (e.g., 'their probability of successfully localizing their prey dropped from XX to approximately 0.35')

In the references, in line 343 italicize '*Canis lupis*'; in line 389 italicize '*Antrozous pallidus*'; in line 403 italicize '*Megaderma lyra*'

In sum, these are minor points that should strengthen an already very strong manuscript, which serves as an excellent contribution to the growing field of the effects of anthropogenic noise on predator foraging behavior.

Referee: 2

Comments to the Author(s)

Passive gleaning bats that rely on prey generated sounds show a decline in hunting success when foraging in noisy environments. The driving mechanism for this decline could either be masking of the prey generated sound by spectral overlap, or distraction of the bat by drawing its attention from the detection and localization task. The aim of this study was to elucidate which mechanism may explain the decline in the bats' foraging performance. In a well-designed experimental approach, the authors trained pallid bats to land on one of 21 speakers broadcasting band-limited rustling noises of insect prey and measured their foraging attempt, foraging success and foraging time in ambient condition and two bandpass limited noise conditions of which one condition spectrally overlapped with the prey cue and the other did not overlap with the cue. The results show a decline in hunting success, an increase in search time and at least in the beginning of the experiment a reduced attempt to land on the speakers in both noise condition, supporting the

distraction hypothesis. In the echolocation behavior bats increased the repetition rate under noise condition, although this had no effect on foraging success. Overall this is a thoroughly conducted experiment which indicates that distraction and not exclusively masking is a primary driver for the reduced foraging performance of passive gleaning bats in noisy environments. This work complements the literature on ambient noise and its effect on foraging success in gleaners and is of high scientific value as it delivers important knowledge on the underlying mechanism. Therefore, this manuscript should be published with some minor revisions (see below).

Minor comments

96f: "Previous work with pallid bat (...), and thereby rescue their foraging behaviors in masked situations" What exactly do you mean? I think this needs a bit more explanation

130: intensity of prey cue is given for 1cm above the speaker, but noise levels are given for the bats' foraging height at 30 cm. Intensity levels should refer to the same distance.

148: Was the background noise in the ambient condition measured at the same position as in the noise condition? The background noise is almost as loud as your prey cue and your noise conditions are only 4.5 dB above the ambient noise level. What was the spectrum of your background noise? I suggest to introduce an own figure showing the sonogram and power spectrum of your three conditions and of the prey cues. The spectrograms in Fig. 1c-e are too small.

195ff: how many calls from how many trials were analysed?

Fig.2: colours, especially yellow, are hardly visible in the printed version. Lines are too narrow and dots too small.

2a and b: I guess, dots indicate single trials. I find this confusing as they are plotted in a graph with a continuous y- axis. At least, it should be mentioned in the figure legend.

2b: y- axis labeling: why "predicted" success? success is clearly defined in the method section as successfully landing on the correct speaker.

288: "We also think it is unlikely" instead of "We also think it unlikely is..."

In the result section you report data on the echolocation behavior (higher repetition rate, no change in peak frequency). These results are not discussed. How do you explain these results? A short paragraph on echolocation behavior should be added in the discussion.

386: Ostwald instead of Ostwalk; J. Exp. Biol. 211, 3174-3180 instead of 315-324

Author's Response to Decision Letter for (RSPB-2020-2689.R0)

See Appendix A.

Decision letter (RSPB-2020-2689.R1)

12-Jan-2021

Dear Dr Allen

I am pleased to inform you that your manuscript entitled "Noise distracts foraging bats" has been accepted for publication in Proceedings B.

Open Access

Your article has been estimated as being 9 pages long. Our Production Office will be able to confirm the exact length at proof stage.

Paper charges

Sincerely,

Dr Sarah Brosnan

Associate Editor:

Board Member

Comments to Author:

(There are no comments.)

Appendix A

Dear Editor,

We thank you and the two anonymous reviewers for the comments on our manuscript entitled "Noise distracts foraging bats". We include the full reviews and respond to comments in detail below (responses are in **bold**). We utilize track changes in the manuscript as requested. We are resubmitting our revised manuscript to Proceedings of the Royal Society B. for further consideration.

Sincerely,
Louise Allen, Jesse Barber, on behalf of all Co-authors

Associate Editor

Board Member: 1

Comments to Author:

Thank you for submitting your manuscript to Proceedings B. We have now received two reviews of your manuscript, both of which express enthusiasm for the elegantly simple experimental design and the broad interest of addressing the mechanism by which acoustically hunting animals are influenced by noise. Both reviewers also provide many valuable comments and suggestions that will improve the clarity of the methods and results in a future version of the manuscript. Reviewer 1, however, expresses concern about the very low sample size and the fact that this is not directly addressed in the manuscript. To be considered for publication in Proceedings B, a revised manuscript would need to effectively address this concern.

We are enthused that the reviewers were positive about our manuscript. We agree that the reviewer's comments were insightful and found that after revision, the manuscript is markedly strengthened.

Reviewer(s)' Comments to Author:

Referee: 1

Comments to the Author(s)

This is an elegant experiment designed to parse out the mechanism by which noise impedes foraging behavior in bats. While a handful of studies have documented a detriment in bat foraging ability in the presence of noise, none have clearly identified the mechanism underlying this decrease in performance. When confronted with noisy surroundings, do bats show reduced foraging performance because they are distracted, or because target sounds are masked?

The study makes a compelling argument and is an excellent contribution to this growing field. My only real concern is the low sample size. The researchers started with five individuals but were unable to proceed through the full experiment with two of them,

leaving a sample size of three. While I find the results very clear (bats showed similar detriment in foraging ability in the presence of both bandwidths of noise, suggesting that it is distraction, not masking, that is impeding foraging performance), I think it is important to stress that individuals can respond to noise (or any environmental disturbance) in different ways, as is nicely evidenced in one of the studies this paper cites (Gomes & Goerlitz 2020). Variation in response is expected in animal behavior, and results from a small number of individuals should be interpreted carefully. I would recommend discussing this sample size limitation explicitly in the discussion in light of Gomes & Goerlitz 2020, perhaps stressing the need for further investigation with more individuals and more species. I think this is a very strong study and an important contribution to the field. But I think some caution should be introduced in the interpretation of the results.

Gomes DGE, Goerlitz HR. 2020 Individual differences show that only some bats can cope with noise-induced masking and distraction. bioRxiv.

Thank you for the kind words. We have added the following text urging caution when interpreting the results from a small sample size and recommend additional work. "... it is important to point out that individuals can respond to environmental disturbances differently [26] and that additional work is necessary with more bats and different species." (lines 341-344).

I have listed some minor comments below:

Line 89-90: I disagree with this definition of gleaning bats: 'those that listen for prey-generated sounds on substrates.' To my understanding, gleaning simply refers to the foraging technique of capturing prey off of substrate (e.g., foliage, the ground). The sensory mode by which gleaning takes place is often attending to prey-generated sounds ('passive gleaning'). But other sensory modalities can be used too. Active gleaners, for example, can actively produce echolocation calls to find silent, motionless prey resting on leaves in the forest understory (e.g., Geipel et al 2013, Proc Roy Soc: Perception of silent and motionless prey on vegetation by echolocation in the gleaning bat, *Micronycteris microtis*). I'd recommend rewording this sentence a bit.

We agree and have changed the wording to "bats who passively glean prey – those that often or sometimes hunt by listening for prey-generated sounds on substrates" to highlight that we are referring to gleaning bats that use prey-generated cues to passively localize their prey (line 95-96).

Line 98: more explanation is needed here to make the meaning clear to the reader: 'Previous work with pallid bats suggests that they cannot use sonar to image small, stationary prey animals, and thereby rescue their foraging behaviors in masked

conditions'. Describe in more detail what is meant by 'rescue' and how this rescue is accomplished.

We agree and have reworded as "Previous work with pallid bats suggests that they cannot use sonar to image small, stationary prey animals [29], and thereby compensate for any deficiency in their ability to passively glean in masked conditions (as in [22])." to clear up this sentence (lines 102-105).

Line 101: add 'we predicted' as follows: 'Alternatively, if distracted we predicted they...'

Good idea. We have made this correction (line 108).

Line 132: Referring to this line in the methods, 'During initial training, we exposed all bats to the full-spectrum cricket walking sounds', what frequency range was this?

We have added the frequency range of the original file (lines 143-144).

Lines 123-133: To limit the bandwidth, did you keep the lower frequency steady but gradually reduce the upper frequency? Perhaps good to specify this here.

We have included the following to address this comment: "During initial training, we exposed all bats to the full-spectrum cricket walking sounds (1.6-17.6 kHz at ± 15 dB, recorded at 10 cm), then gradually limited the bandwidth of the prey cue by contracting the upper and lower frequency bounds until the bat was unable to localize the sound source." (lines 142-145).

Line 137: change 'signal' to 'cue'

We have made this change.

Line 189: change 'cries' to 'calls'

We have made this change.

Line 283: I would be more cautious in the classification of the foraging mode of *Phyllostomus hastatus*. While *P. hastatus* does indeed consume many insect prey (as well as flexibly and opportunistically consuming small vertebrates, fruits and flowers), to my knowledge it has not been shown how it catches its insect prey. And while it likely gleans prey from substrates, it is unclear whether it uses prey-produced sounds to do

so. More studies are needed on this bat... Maybe change 'a bat that also gleans prey' (line 283) to 'a bat that also consumes large insects'.

We have made this change (line 363).

Line 294: It would be nice to have more of a discussion of what might be going on during distraction. The authors elude to a reduction in attention or other cognitive processes, and a decrease in processing power. Can you go into a little more depth on what attentional/cognitive processes might be impeded in distraction? What studies have looked at this and what might be done next to investigate these processes in bats?

We have added the following short paragraph to better address distraction. "Our findings supporting distraction as a primary mechanism driving hunting deficits have important implications. Distraction does not depend on the overlap in properties between the noise and the stimulus. Thus, distraction can occur within and across sensory modalities. Future work should focus on higher-level cognitive processes, such as spatial orientation [47] and memory retrieval [48]. This work would benefit from using the human literature as a guide to form hypotheses for non-human animals. For instance, data indicates that noise interferes with learning and problem-solving capacity in humans, likely due to distraction or related phenomena[49]" (lines 432-438).

Line 311: What was the hunting efficacy of the bats without noise? Maybe good to add that here to give perspective (e.g., 'their probability of successfully localizing their prey dropped from XX to approximately 0.35)

Excellent point. We have made the following addition "probability of successfully localizing their prey dropped from 0.94 (CI: 0.89-0.98) in ambient conditions to approximately 0.35 (CI: ~0.25-0.50) in both noise treatments" (lines 447-448).

In the references, in line 343 italicize 'Canis lupis'; in line 389 italicize 'Antrozous pallidus'; in line 403 italicize 'Megaderma lyra'

Thank you for catching these! We have made your suggested changes.

In sum, these are minor points that should strengthen an already very strong manuscript, which serves as an excellent contribution to the growing field of the effects of anthropogenic noise on predator foraging behavior.

Thank you again for the kind words and thoughtful feedback.

Referee: 2

Comments to the Author(s)

Passive gleaning bats that rely on prey generated sounds show a decline in hunting success when foraging in noisy environments. The driving mechanism for this decline could either be masking of the prey generated sound by spectral overlap, or distraction of the bat by drawing its attention from the detection and localization task. The aim of this study was to elucidate which mechanism may explain the decline in the bats' foraging performance. In a well-designed experimental approach, the authors trained pallid bats to land on one of 21 speakers broadcasting band-limited rustling noises of insect prey and measured their foraging attempt, foraging success and foraging time in ambient condition and two bandpass limited noise conditions of which one condition spectrally overlapped with the prey cue and the other did not overlap with the cue. The results show a decline in hunting success, an increase in search time and at least in the beginning of the experiment a reduced attempt to land on the speakers in both noise condition, supporting the distraction hypothesis. In the echolocation behavior bats increased the repetition rate under noise condition, although this had no effect on foraging success. Overall this is a thoroughly conducted experiment which indicates that distraction and not exclusively masking is a primary driver for the reduced foraging performance of passive gleaning bats in noisy environments. This work complements the literature on ambient noise and its effect on foraging success in gleaners and is of high scientific value as it delivers important knowledge on the underlying mechanism. Therefore, this manuscript should be published with some minor revisions (see below).

Thank you for the insightful comments. The manuscript is decidedly stronger after incorporating your feedback.

Minor comments

96f: "Previous work with pallid bat (...), and thereby rescue their foraging behaviors in masked situations" What exactly do you mean? I think this needs a bit more explanation

We have reworded as "Previous work with pallid bats suggests that they cannot use sonar to image small, stationary prey animals [29], and thereby compensate for any deficiency in their ability to passively glean in masked conditions (as in [22])." to clear up this sentence (lines 102-105).

130: intensity of prey cue is given for 1cm above the speaker, but noise levels are given for the bats' foraging height at 30 cm. Intensity levels should refer to the same distance.

We have added this information.

148: Was the background noise in the ambient condition measured at the same position

as in the noise condition? The background noise is almost as loud as your prey cue and your noise conditions are only 4.5 dB above the ambient noise level. What was the spectrum of your background noise? I suggest to introduce an own figure showing the sonogram and power spectrum of your three conditions and of the prey cues. The spectrograms in Fig. 1c-e are too small.

We have added a spectrogram of the prey cue to Figure 1 to provide additional clarity. We describe this in our figure caption (lines 624-628). Power spectra of the noise stimuli would not be informative as we used bandpassed white noise with equal energy across frequencies. We have also added a sentence to the figure caption to clarify this for our readers (lines 628-629).

195ff: how many calls from how many trials were analysed?

Thank you for pointing this out! We have added a sentence on lines 258-259 and modified the sentence on line 260-262.

Fig.2: colours, especially yellow, are hardly visible in the printed version. Lines are too narrow and dots too small.

We agree and have made the color for the non-overlapping treatment darker and have thickened the lines around the boxplots and trendlines, as well as increased the size of the data points.

2a and b: I guess, dots indicate single trials. I find this confusing as they are plotted in a graph with a continuous y- axis. At least, it should be mentioned in the figure legend.

We have revised the last sentence of the figure legend to be more clear (lines 639-641).

2b: y- axis labeling: why “predicted” success? success is clearly defined in the method section as successfully landing on the correct speaker.

We have modified the figure axis to just say “success”. This initially said predicted because our visualization is based on modeled data, but it is perfectly acceptable to simply refer to this as success.

288: “We also think it is unlikely” instead of “We also think it unlikely is...”
In the result section you report data on the echolocation behavior (higher repetition rate, no change in peak frequency). These results are not discussed. How do you explain

these results? A short paragraph on echolocation behavior should be added in the discussion.

Thank you for catching this! We have included a short paragraph on echolocation behavior as suggested (lines 337-344).

386: Ostwald instead of Ostwalk; J. Exp. Biol. 211, 3174-3180 instead of 315-324

We have made this change.